# The Effects of Daily Temperature on Crime Events in Urban Hanoi, Vietnam Using Seven Years of Data (2013–2019)

**DOI:** 10.3390/ijerph192113906

**Published:** 2022-10-26

**Authors:** Vu Thuy Huong Le, Jesse D. Berman, Quynh Anh Tran, Elizabeth V. Wattenberg, Bruce H. Alexander

**Affiliations:** 1Department of Environmental Health Sciences, School of Public Health, University of Minnesota, Minneapolis, MN 55455, USA; 2Department of Environmental Health, School of Preventive Medicine and Public Health, Hanoi Medical University, Hanoi 10000, Vietnam

**Keywords:** crime, climate change, temperature, hot weather, Vietnam, violent crime, violent behavior, developing countries

## Abstract

The effects of temperature on behavior change and mental health have previously been explored, but the association between temperature and crime is less well understood, especially in developing countries. Single-city-level data were used to evaluate the association between the short-term effects of temperature on crime events in urban Hanoi, Vietnam. We used quasi-Poisson regression models to investigate the linear effects and distributed lag non-linear models to investigate the non-linear association between daily temperature and daily crime events from 2013 to 2019. There were 3884 crime events, including 1083 violent crimes and 2801 non-violent crimes during the 7-year study period. For both linear and non-linear effects, there were positive associations between an increase in daily temperature and crime, and the greatest effects were observed on the first day of exposure (lag 0). For linear effects, we estimated that each 5 °C increase in daily mean temperature was associated with a 9.9% (95%CI: 0.2; 20.5), 6.8% (95%CI: 0.6; 13.5), and 7.5% (95%CI: 2.3; 13.2) increase in the risk of violent, non-violent, and total crime, respectively. For non-linear effects, however, the crime risk plateaued at 30 °C and decreased at higher exposures, which presented an inverted U-shape response with a large statistical uncertainty.

## 1. Introduction

Ambient temperature is linked to several human health effects, including communicable and non-communicable diseases, and behavior changes. Whereas a large body of the epidemiologic literature describes the relationship of temperature with mortality, morbidity, and hospitalization [1,2], fewer studies have been conducted to assess the relationship between temperature and criminal behavior. Recent evaluations have shown the association between daily temperature and crime events, with most focused on developed countries. For example, a study in Finland showed that a 2 °C increase in average temperature increased violent crime rates by more than 3% [3]. Research across 436 U.S. counties estimated that each 10 °C increase in daily temperature was associated with an 11.92% increase in the risk of violent crime and a 6.14% increase in the risk of non-violent crime [4]. A study in one U.S. city (Philadelphia, Pennsylvania) observed the highest crime rates during a mean daily heat index of 22.6–28 °C [5].

Little research has been conducted in developing countries to examine the effects of temperature on crime, and no study has been conducted in Vietnam. Gates et al. found an increase in homicides for each 1 °C in maximum temperature in South Africa [6]. However, Trujillo et al. indicated that each 1 °C in maximum temperature is associated with a 0.8% increase in interpersonal violence but decrease in homicides in Colombia [7]. Existing research found that adverse weather shocks in the form of droughts lead to an increase in violent crime in rural Brazil [8]. A single-city study in China showed positive associations between temperature and both violent and non-violent crimes [9]. These studies indicate that ambient temperature is associated with crime rates, but different regions may show different relationships.

Related to this problem is the larger issue of climate change, which poses significant public health concerns, threatening humanity through the perpetuation of adverse environmental consequences, such as extreme weather events, and wildfires, and increased temperature [10]. Accordingly, temperature-related diseases, including violent behavior, are matters of increasing public health significance.

To the best of our knowledge, no study has evaluated the association between temperature and crime events in Hanoi, Vietnam. Vietnam is a Southeast Asian nation, a region at high risk for climate change-related hazards like rising temperature [11]. Our investigation examines the potential temperature–crime associations in this understudied region, using seven years of data from 2013 to 2019 to examine our hypothesis that an increase in daily temperature will be associated with rising daily crime events in urban Hanoi.

## 2. Materials and Methods

We conducted an ecological study to estimate the short-term association between temperature and the risk of criminal behavior in urban Hanoi, Vietnam. We collected data on daily temperatures and crime events from 2013 to 2019. We considered two types of crime, violent and non-violent crime, and three temperature measures, daily minimum temperature, daily mean temperature, and daily maximum temperature.

### 2.1. Study Area

Hanoi is the capital of Vietnam and is in a sub-tropical climate region. Based on the Annual National Population and Household Report 2019, Hanoi covered an area of 3358 km^2^ and an estimated density of 2239 people/km^2^ (https://www.gso.gov.vn/dan-so/ accessed on 6 June 2022). Over ten years the annual population growth has been approximately 4.4% in urban areas, compared to 0.72% in rural areas. Hanoi is divided into 30 districts, including 12 urban districts, one district-leveled town, and 17 rural districts. There are four seasons in Hanoi, Spring (Feb–April), Summer (May–July), Fall (Aug–Oct), and Winter (Nov–Jan), with the lowest temperature in January and the highest temperature in June or July [12].

### 2.2. Crime Count Data

The study outcome was daily reported crime counts. All criminal activities were coded with the Vietnamese Criminal Code (No 100/2015/QH13). We obtained crime data from 1 January 2013 to 31 December 2019 based on daily reports in the PC02 Department, Police Headquarters Hanoi.

The PC02 Department reported six common types of crime representing violent behavior that might physically harm an individual through injury, force, or threat of injury, and three common types of crime representing non-violent behavior that is directed at an individual’s property. The six violent crime types included murder/manslaughter, rape, assault, lewd and being lascivious behavior with a child under 16, kidnapping, and robbery. Robbery was in the violent crime group because it is defined as “Any person who uses violence, the threat of immediate violence, or commits other acts that render another person unable to resist to obtain his/her property”. Three non-violent crimes were: larceny, burglary, and fraud. Other non-violent crimes that are not directed at individuals, such as drug-related and prostitution, were not included in this study because those types of crime were not reported and managed by the PC02 Department.

Each type of daily criminal activity is reported by the local District Police Office and aggregated by Police Headquarters Hanoi using paper reports. The reports include the name of the criminal, age, type of crime, time, date, and location where the crime was committed. For the purposes of this study, only the date, time, location, and type of crime were ascertained. Information on age, sex, personal characteristics, or identifying information were not allowed to be collected for this study in accordance with Vietnamese national security protocols.

A total of 1826 paper reports were collected from 2013 to 2019. The assistance of police department personnel was obtained to enter data from paper reports using an Epidata Software v4.6.0.2 (http://www.epidata.dk/ accessed on 25 December 2021). A 10 percent random sample of the data was checked to ensure the validity of the data entry process; this determined a 99% accuracy rate.

### 2.3. Temperature and Covariates

The Hanoi Environment and Natural Resource Department and Vietnam Environment Administration maintains 11 weather station monitors, including three stationary monitors and eight mobile monitors, that report hourly atmospheric data, including temperature, fine particulate matter (PM_2.5_), and relative humidity (R.H.).

Estimates of daily environmental measures based on hourly data were calculated in several steps. First, we identified daily minimum, maximum, and mean temperature based on the hourly temperature data of each air monitor. Illogical measurements and outliners were removed. In each air monitor, we removed all days with daily maximum temperatures below daily minimum temperature, while removing all values under 5 °C for daily minimum temperature, over 44 °C for daily maximum temperature, and under 7 °C or over 40 °C for daily mean temperature. These cutpoints were based on the lowest and highest value of temperature measurements in Hanoi reported in the General Statistics Office of Vietnam (https://www.gso.gov.vn/en/homepage/ accessed on 15 January 2022) and a previous study [13]. Finally, we calculated an average of each daily temperature measurement from the valid daily air monitors. Appendix A displays characteristics of daily temperature measurements in each air monitor. A pairwise correlation across the 11 monitors showed strong and positive correlations, ranging from 0.87 to 0.99 (Appendix A).

This study considered day of the week, holidays, season, year, and additional atmospheric conditions (PM_2.5_ and R.H.) as time-variant confounders of crime. Holiday was a binary indicator for whether a day was classified as holiday or non-holiday. Holiday included all Vietnamese public holidays, including New Year’s Eve, Lunar New Year’s Event, Labor Day, Independence Day, and the preceding Friday or following Monday if a holiday fell on a Saturday or Sunday, respectively. (https://www.officeholidays.com/countries/vietnam/2020 accessed on 6 April 2021). Daily PM_2.5_ (μg/m^3^) and R.H. (%) represents the average of daily mean measurement of the 11 station monitors.

### 2.4. Statistical Analysis for Linear Effects

We estimated the risks of daily variation in crime counts associated with daily variation in temperature with a quasi-Poisson family for overdispersed data, adjusting for long-term effects of year and seasonal trends (Winter, Spring, Summer, and Autumn). The results were reported as the percent increase in the relative risk (RRI) of a crime occurrence for each 5 °C increase in daily minimum temperature, mean temperature and maximum temperature, separately. The interpretation of the RRI can be considered the percent relative effect and is directly derived from the relative risk. Reporting an RRI allows for a change in risk that may be lower in magnitude, but still statistically significant, to be more easily interpreted. This is calculated as the (Relative Risk −1) × 100 [4].

The primary model (Model 1) was adjusted for day of week and holiday status. Model 2 was adjusted for day of week, holiday status and daily relative humidity. Model 3 was adjusted for day of week, holiday status and daily PM_2.5_. Model 4 was adjusted for day of week, holiday status, daily relative humidity and daily PM_2.5_. We estimated the RRI at seven individual lag day models, from 0 to 7 days, to identify the delayed effects of temperature and crime counts. All models were also fit separately for each temperature measurement and crime outcome: violent crime, non-violent crime, and the combination of violent and non-violent crime.

### 2.5. Statistical Analysis for Non-Linear Effects

Distributed lag non-linear models (DLNM) were fit to estimate the relative risk of each 1 °C increase in daily temperature and each type of crime. A natural cubic spline with 4 degrees of freedom and 7 lag days for temperature was applied in the DLNM. The modelling framework can describe non-linear relationships both in the day of exposure and lags [14]. The models were adjusted for potential time-varying confounders of crime, including the day of the week and holiday status. Season and year were used to control for seasonal and long-term trends. The results were interpreted as the effect of the exposure versus a reference and reported as both distributed lag effect and individual lag days [15]. The choice of the centering value depends on the interpretational issues and does not affect the fit of the model [16]. The curve was centered at 12 °C, which represents the 1^st^ percentile of daily mean temperature for the entire 7 years to demonstrate the change in risk from a low-temperature exposure point. The minimum recorded temperature was not selected to reduce the effects of extreme outliers in the final model.

Analyses were performed using R Software analysis, version 4.0.5, R Foundation for Statistical Computing (Vienna, Austria) relying on the dlnm package for statistical assessments [14]

### 2.6. Sensitivity Analysis

We evaluated the sensitivity of the models to different elements: (1) estimating the linear effects of temperature and crime using separated models stratified by holiday status (holiday and non-holiday) and weekend status (weekend and weekday), (2) applying times series analysis adjusted for a time component to control for seasonal and long-term trends using natural cubic spline with 3, 5, 7 and 9 degrees of freedom per year for linear effects.

## 3. Results

We identified 3884 total crimes, including 1083 violent crimes and 2801 non-violent crimes across 2556 data days from 1 January 2013 through 31 December 2019. Crime incidence changed seasonally with more crime occurring in Spring (January through April; Appendix A). Appendix A shows the violent and non-violent crime counts per year, with the highest observed crime counts in 2014 and a decreasing trend in subsequent years.

Mean daily minimum, mean, and maximum temperatures were 22.8 °C, 25.8 °C and 30.1 °C, respectively (Table 1). On average, there were 1.52 crime occurrences per day with 0.43 daily violent crimes and 1.10 daily non-violent crimes. The average monthly crime counts fluctuated, with the highest counts observed in April, May, and September (Appendix A). Those months also had the highest number of public holidays in Vietnam. Appendix A shows a strong observed correlation between three types of daily temperature measures. Negative correlations were observed between temperature measures and relative humidity and PM_2.5_ (Appendix A). The highest daily mean temperature was observed in June and July, and the lowest daily mean temperature was in January (Appendix A). The standard deviations of daily mean temperature in winter (November to January) were greater than in summer (May to July). Appendix A shows the distribution of daily crime counts by type and the presence of several days without any crime events.

The estimated RRI revealed that each 5 °C increase in daily minimum, mean and maximum temperature was positively associated with violent crime, non-violent crime, and all types of crime counts in the city of Hanoi (Table 2). The association between all temperature measures and non-violent crime and all crime types were similar across adjusted and unadjusted models, with the strongest association for violent crime. The crime risks were consistently higher for mean temperature compared to risks from minimum and maximum temperature, which may reflect the mean being a better characterization of the effects of temperature on behavior. Additionally, models employing daily mean temperature gave a slightly lower AIC value, showing some evidence of a better model fit (described for Model 1 in Appendix A). Accordingly, we believe the mean temperature is the most representative for this analysis.

The observed association between daily mean temperature and crime was the greatest on the day the crime was committed (Lag 0), with highest risk in violent crime (RRI = 9.9, 95%CI = 0.2; 20.5), followed by all crime types (RRI = 7.5, 95%PI = 2.3; 13.2) and non-violent crime (RRI = 6.8, 95%CI = 0.6; 13.5) (Table 3). The crime–temperature associations decreased with each daily lag, and null effects were observed after Lag 0 for non-violent and violent crime, but with Lag 3 for total crimes (Table 3).

In our non-linear distributed lag model, we observed an increase in the risk of violent crime, non-violent crime, and total crime with increased daily mean temperature until around 30 °C, at which point the risk begins to decline in an inverted U-shape response (Figure 1). At a daily mean temperature of 30 °C, the risk of violent crime (RR = 1.8, 95%CI = 0.9, 3.6) was similar to that for non-violent crime (RR = 1.7, 95%CI = 1.1, 3.6). Confidence intervals reveal a wide range of uncertainty in the non-linear estimates.

Figure 2 demonstrates that the individual lag curves for 20 °C, 30 °C, and 35 °C presented the greatest crime risks on the first day of exposure, then the risks decreased on the following days (lag 1 to lag 7), with the reference at 12 °C. The increased risk of violent crime was greater in magnitude compared to non-violent crime. The relative risk for violent crime was lowest at 20 °C and highest at 30 °C but decreased at 35 °C. The relative risk for non-violent crime was similar in magnitude at the three different temperatures.

Appendix A show the individual lag–response curves for violent crime, non-violent crime, and total crime, respectively. The figures confirm that an increase in mean daily temperature increased the risk of crime on the first day of exposure, and particularly for violent crime.

Crime counts in Hanoi were characterized by relatively few events per day without a sharp seasonal pattern observed. We tested the sensitivity of our model to two temporal smoothers: the year and season variable and a natural cubic spline model for time (Appendix A). It was observed that the spline model resulted in poor model fit with high sensitivity towards degrees of freedom and attenuated effect estimates. Therefore, the season–year metric was identified as a preferred temporal smoother.

## 4. Discussion

In this study we observed an association between a short-term increase in daily temperature and an increased risk of violent and non-violent crime in Hanoi, Vietnam. The greatest effect was observed on the day of exposure, and the association between crime and mean temperature was non-linear, with an increased risk up to 30 °C, which then decreased at higher temperatures, supporting previous research in other populations [4]. Our results were robust to metrics of maximum, mean, and minimum daily temperatures with similar associations. To our knowledge, this is the first study to investigate an association between daily temperature and crime risk in Vietnam.

Urban Hanoi offers a unique population to explore temperature–crime associations. Hanoi is in northern Vietnam which has a sub-tropical climate with four seasons per year. This differs from cities in the southern regions of Vietnam, such as Ho Chi Minh city, where two seasons are characterized as dry or rainy. Therefore, the temperature–crime associations in Hanoi may differ from Ho Chi Minh city, as well as cities in the Americas, Africa, and Europe. Hanoi is the second-largest city in Vietnam and has experienced rapid growth through urbanization in the last two decades. Hanoi’s urban structure and boundary were expanded in 2008, and policies have strongly promoted population density and social development [12]. Moreover, in contrast to other urban areas in which studies of temperature and crime have been conducted, our data indicate that Hanoi has much lower crime rates than urban areas in other parts of the world, with an average crime count of 1.52 per day, compared to 39.7 in South Africa or 135 in Baltimore, Maryland [6,17], although this may be partially attributable to differences in policing and crime reporting.

There is an increased interest in whether the rising temperatures could affect criminal behavior [4,6,17,18]. One study reported a significant and inverted U-shaped association between daily temperature and crime risk across 436 U.S. counties for multiple cities level. Each 10 °C increase in daily temperature was associated with an increased the risk of violent crime and non-violent crime [4]. Another study, focused on intentional crime, showed linear effects, with each 5 °C in daily mean temperature leading to a 9.5% increase in crime risk [18]. A study conducted in South Africa indicated a one-degree Celsius increase in daily maximum temperature was associated with a 1.5% increase in crime risk [6]. Our finding shows a similar trend and magnitude, but with a reduced precision that is likely a product of the reduced crime events from our single low-crime city study. Other single-city studies have also identified positive relationships between temperature and criminal behavior [3,5,17]. For example, a study in Baltimore, Maryland reported an association between daily maximum temperature and increased total and violent crime [17], but did not examine daily mean or minimum temperature.

A limitation of single-city analyses is that they cannot be generalized to larger geographic regions due to the different social, demographic and climate factors. Our study was conducted in Hanoi, located in the north of Vietnam, so it may not be generalizable to elsewhere in the country. However, the results from a single-city study can raise the public’s awareness of climate change at the local level.

This study contributed evidence of a potential linear and non-linear temperature-crime association. For linear effects, we observed an increase in crime risk for each 5 °C increase in temperature, in line with other single-city studies in South Africa and Maryland. The non-linear effect of daily mean temperature showed an inverted U-shaped response with a downward deflection at a very high temperature. This can be explained by the Social Escape or Avoidance Theory and the Negative Affect Escape Model [19,20], and is supported by previous observational studies [4,18]. At very high temperatures, people will reduce social interaction to avoid heat, so crime risk may decrease. This finding may have implications for potential effects of future heatwave events on criminal behavior, which requires further investigation.

We used three temperature measures to estimate temperature–crime associations. Daily minimum, mean and maximum temperature show similar associations and trends, but the precision varied across five models when adjusted for various confounding factors. Several studies have used one or two temperature measurements to examine the effects of temperature on crime, such as minimum and maximum daily temperature or only mean daily temperature [17,21,22]. Different temperature exposures can highlight the exposure at different times of day (e.g., maximum temperature is indicative of daytime temperatures, while minimum temperature is indicative of early morning temperatures), which may be important for our behavior-related outcome. Our study found that minimum, maximum, and mean temperatures had a similar predictive ability because of their strong correlation. We primarily used the mean temperature as a metric as it was the more statistically efficient model based on the AIC, but this critically demonstrated that all temperature conditions can be useful for evaluating criminal behavior.

Several limitations need to be acknowledged in this study. First, individual-level information about people who commit crimes was not obtainable due to policy use restrictions on the available data. Thus, we were not able to examine differences in risk by age, gender, or other factors. However, our study investigated the risks in a single population where the underlying distribution of demographic factors is unlikely to change across short time durations, so these factors would be unlikely to confound the population-level results. Second, the original data reported to the Hanoi Police Department were recorded on a paper report, which was then abstracted to an electronic record. This process poses a risk of errors during the data-entry process, but we believe that these were minimized by the data-entry verification processes established in the abstraction process. Third, many crimes are underreported for various reasons, so the absolute crime burden is not known. Additionally, some crimes, such as drug use and prostitution, were not reported by the PC02 Department with other violent/non-violent crime types. The extent to which this underreporting is related to temperature is unknown, and there may be some selection bias that could not be accounted for in our analyses. Fourth, this study was limited to seven years, which, in concert with the relatively low rate of crime in Hanoi, resulted in a small number of crime events compared to other studies, thus limiting the precision of our estimates. The small number of events limited our ability to adjust for the time variable because under- or over-smoothed estimations may occur based on the chosen number of degrees of freedom per year [23,24]. Additionally, a number of climate factors have been found to be correlated with crime, such as ozone, wind speed, and precipitation [21,25,26], but were not available for this study. Finally, it should be emphasized that our study explored a single city and our findings may not be generalizable to other cities in Vietnam or Southeast Asia.

## 5. Conclusions

The observed association between temperature and crime may be of particular concern in the context of climate change, especially in low- and middle-income countries. To the best of our knowledge, this study is the first study in Vietnam to examine the relationship between short-term temperatures and criminal behavior. Our findings from both linear and non-linear models support the hypothesis that temperature influences criminal behavior. We expect that our results may be valuable for local police planning and social services in urban Hanoi, Vietnam. Our results indicate that additional investigations that including other regions of Vietnam are warranted.

## Figures and Tables

**Figure 1 ijerph-19-13906-f001:**
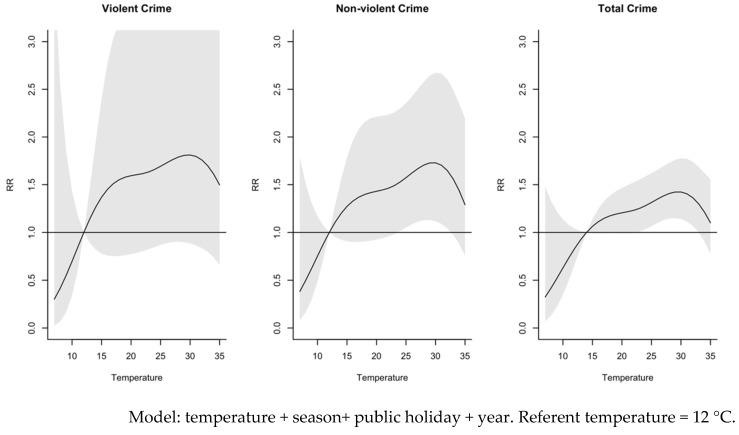
Overall exposure-response associations between daily mean temperature and violent crime (**left**), non-violent crime (**middle**), and total crime (**right**) in urban Hanoi, Vietnam, 2013–2019. The models represent distributed lag curves and shaded areas denote 95% confidence intervals.

**Figure 2 ijerph-19-13906-f002:**
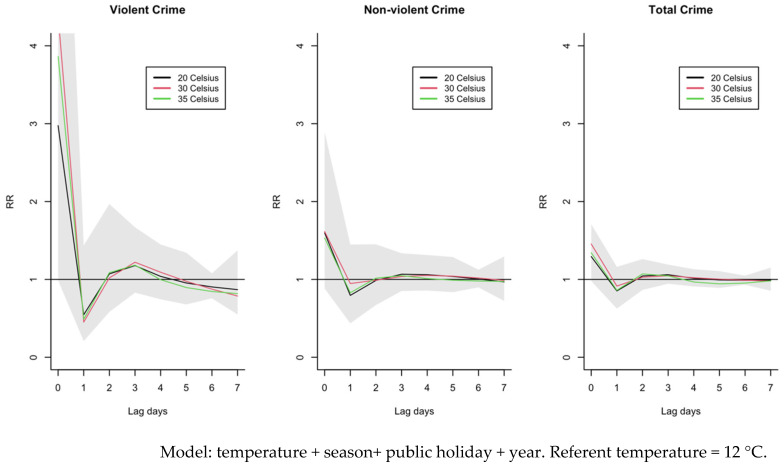
Individual lag–response curves for different temperatures (at 20 °C, 30 °C and 35 °C), reference 12 °C in urban Hanoi, Vietnam, 2013–2019. The models represent individual lag curves (from 0 to 7 days) and shaded areas denote 95% confidence intervals.

**Table 1 ijerph-19-13906-t001:** Daily weather conditions and crime occurrence by type in urban Hanoi, Vietnam, 2013–2019.

	Minimum	25th	Median	75th	Maximum	Mean	SD
Minimum temperature (°C)	5.8	19.5	24.1	26.5	33.3	22.8	4.9
Mean temperature (°C)	6.9	21.7	26.9	30.2	38.5	25.8	5.6
Maximum temperature (°C)	8.2	25.3	31.3	35.5	42.6	30.1	6.7
Relative humidity (%)	30.2	66.5	74.3	82.5	99.9	73.9	12.5
PM_2.5_ (μg/m^3^)	0.33	28.3	37.6	52.8	197.1	44.7	26.8
Violent crime	0	0	0	1	5	0.43	0.68
Non-violent crime	0	0	1	2	8	1.10	1.22
All types of crime	0	0	1.5	2	11	1.52	1.42

25th and 75th refer to the 25th and 75th percentile values daily weather conditions and crime occurrence by types.

**Table 2 ijerph-19-13906-t002:** Crude and adjusted models of estimated relative risk increase for violent and non-violent crime for each 5 °C increase in daily mean, maximum and minimum temperature on the day of exposure in urban Hanoi, Vietnam, 2013–2019. An asterisk denotes statistically significant estimates (*p* < 0.05).

	Model 0 **	Model 1 ^+^	Model 2 ^±^	Model 3 ^¶^	Model 4 ^§^
	RRI (95%CI)	RRI (95%CI)	RRI (95%CI)	RRI (95%CI)	RRI (95%CI)
Violent Crime					
Minimum	7.8 (−2.5; 19.1)	7.1 (−3.9; 18.3)	4.0 (−7.7; 15.5)	9.9 (−1.3; 22.5)	7.7 (−4.0; 20.9)
Mean	10.4 * (0.7; 21.1)	9.9 * (0.2; 20.5)	7.9 (−2; 18.9)	11.7 * (1.2; 23.2)	10.2 (−0.8; 22.5)
Maximum	8.1 * (0.6; 16.2)	7.9 * (0.4; 15.9)	6.2 (−2.2; 14.8)	8.9 * (0.9; 17.5)	7.2 (−1.4; 16.6)
Non-violent Crime					
Minimum	5.9 (−0.1; 13.1)	5.6 (−1.0; 12.7)	6.3 (−0.1; 14.1)	7.0 (−0.4; 14.9)	8.4 * (0.3; 17.2)
Mean	7.08 * (0.7; 13.7)	6.8 * (0.6; 13.5)	7.7 * (1.0; 15.0)	8.1 * (1.3; 15.4)	10.0 * (2.5; 18.2)
Maximum	5.05 * (0.2; 10.1)	5.0 * (0.2; 10.1)	5.9 * (0.6; 11.6)	5.6 * (0.4; 11.1)	7.4 * (1.5; 13.6)
All Crime types					
Minimum	6.3 * (0.6; 12.2)	5.9 * (0.3; 11.8)	5.6 (−0.1; 12.0)	7.6 * (1.4; 14.2)	8.2 * (1.4; 15.4)
Mean	7.8 * (2.5; 13.3)	7.5 * (2.3; 13.2)	7.8 * (2.1; 13.7)	8.9 * (3.2; 15.0)	10.1 * (3.7; 16.8)
Maximum	5.8 * (1.7; 10.1)	5.7 * (1.7; 10.0)	6.0 * (1.5; 10.7)	6.5 * (2.0; 11.0)	7.3 * (2.4; 12.5)
Missing data (%)	32 (1.25%)	32 (1.25%)	281 (10.99%)	309 (12.10%)	544 (21.28%)

*: *p* < 0.05; ** Base Model: temperature + season+ year; ^+^ Model 1: Base + public holiday + day of week; ^±^ Model 2: Model 1 + Relative humidity; ^¶^ Model 3: Model 1 + PM_2.5_; ^§^ Model 4: Model 1 + Relative humidity + PM_2.5_.

**Table 3 ijerph-19-13906-t003:** Estimated percent change in the risk of committing crime for each 5 °C increase in daily mean temperature by lag and crime type with 95%CI in urban Hanoi, Vietnam, 2013–2019.

Lag Day	Violent Crime		Non-Violent Crime		Total	
	RRI **	95%CI	RRI *	95%CI	RRI *	95%CI
Lag 0	9.9 *	0.2; 20.5	6.8 *	0.6; 13.5	7.5 *	2.3; 13.2
Lag 1	5.6	−3.6; 15.6	5.8	−0.4; 12.3	5.7 *	0.5; 11.1
Lag 2	4.7	−4.4; 14.6	5.6	−0.5; 12.1	5.3 *	0.2; 10.7
Lag 3	4.4	−4.4; 14.3	4.4	−1.7; 10.8	4.3	−0.8; 9.6
Lag 4	2.1	−6.6; 11.7	3.3	−2.7; 9.6	2.9	−2.1; 8.1
Lag 5	1.4	−7.2; 10.9	4.2	−1.8; 10.5	3.3	−1.6; 8.6
Lag 6	0.5	−9.1; 9.8	2.1	−3.8; 8.3	1.6	−3.2; 6.7
Lag 7	0.1	−8.4; 9.3	2.9	−3.0; 9.1	2.0	−2.8; 7.2

* *p* < 0.05; ** Model: temperature + season+ public holiday + year.

## Data Availability

Not applicable.

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
