# Peer review of "The Effects of Daily Temperature on Crime Events in Urban Hanoi, Vietnam Using Seven Years of Data (2013–2019)"

_ijerph, 2022, doi:10.3390/ijerph192113906_

Round 1

Author Response

Dear Reviewer,

We thank the Reviewer for your comments regarding our manuscript. We have made several changes based on the Reviewer’ comments to improve the manuscript.

First, the structure of the articles is not clear, the discussion of temperature and crime in other countries should be placed in the introduction. And the title “outcome” should be placed by “data description”

We thank the Reviewer for this comment. In the Introduction we have added additional information about previous research in other countries. See paragraph two (Lines 48 - 55). In the discussion, we compare our results to existing research and discuss the different or similar results.

Additionally, we have made some changes to the sub-headings in the Methods

“2.2. Crime count data” (Line 86)

“2.3. Temperature and covariates” (Line 115)

Secondly, for the classification of violent crime and non-violent crime, temperature is not considered as the classification condition of crime types. Whether the temperature has an impact on each type of crime, the article has no basis to support the general use of violence and non-violence to divide crime types

We thank the Reviewer for this comment. We changed the two words “violent behaviors” to “behavior changes” (Line 37) to better describe the effects of temperature on human behaviors. Moreover, we found that temperature affects crime differently. For example, recent studies showed the associations between temperature and violent crimes, but others showed the associations between both violent and non-violent crimes. It is the reason why we examined the effects of temperature on two types of crimes, and we also combine two crimes together. The use of the terminology of violent and non-violent crimes follows previous definitions that violent crimes are directed against an individual with the intent to harm and non-violent crimes are directed at property. We have clarified this distinction in lines 91 - 94

“The PC02 Department reported six common types of crime representing violent behavior, that which might physically harm an individual, and three common types of crime representing non-violent behavior, that which us directed at an individual’s property” (Lines 91-94)

Finally, I think the following specific shortcomings must be fixed:

(1) In title 2.3, 2.4, 2.5, I think some formula explanations can be added to description of the model in the experimental method part. And the definition and calculation of RRI, RR and relative risk of crime are not explained

We thank Reviewer for this comment. There are 5 formulas in this manuscript, and we added them to the Supplement to help audiences follow the model structure.

The Relative Risk (RR) is our standard interpretation of the transformed estimates from the adjusted main model. It denotes the probability of a crime event occurring for each unit increase in temperature. A Relative Risk Increase (RRI) is merely a transformation of the RR, whereby we report it as a percent change in the probability of a crime event occurring for each unit increase in temperature. For example, if an estimated RR was 1.035, this could be interpreted as a 3.5% RRI. Regardless of the interpretation, the conclusions of our model remain the same. Below is modified text to help the readers better understand the RRI.

The interpretation of the RRI can be considered the percent relative effect and is directly derived from the relative risk. Reporting the RRI allows a change in risk that may be lower in magnitude, but still statistically significant, to be more easily interpreted. It is calculated as the (Relative Risk -1) x 100[4]” (Lines 148 - 151)

(2) The format of the table shall indicate the significance of the abscissa and ordinate. For example, in Table 1, the contents in the first row are not explained, such as 25% and 75%

We thank Reviewer for this comment. We added changes in text and made a not under the Table 1

Minimum

25th

Median

75th

Maximum

Mean

SD

Minimum temperature (°C)

5.8

19.5

24.1

26.5

33.3

22.8

4.9

“25th and 75th indicated the 25th and 75th percentile values daily weather conditions and crime occurrence by types” (Lines 206-207)

(3) Since there are many papers on temperature and the number of crimes, the innovation of the articles should be highlighted in the discussion. There is also little innovation in methods. Besides, in line 267- line 280 shows that there have been articles on temperature and crime, which contradicts the fact that line 37-38 here says that there is less research on temperature and crime.

We thank the Reviewer for this suggestion. We added text to highlight the innovation of the articles in the introduction, including the importance of perming this work in an environment of southeast Asia.

“Little research has been done in developing countries to examine the effects of temperature on crime, and no study has been done in Vietnam. Gates et al., found an increase in homicides for each 1 °C in maximum temperature in South Africa[6]. However, Trujillo et al., indicated that each 1 °C in maximum temperature is associated with 0.8% increase in interpersonal violence but decrease in homicides in Colombia[7]. Existing research found that adverse weather shocks in the form of droughts lead to an increase in violent crime in rural Brazil[8]. These studies indicate that ambient temperature is associated with crime rates, but different regions may show different relationships.

Related to this problem is the larger issue of climate change, which poses significant public health concerns, threatening humanity through the perpetuation of adverse environmental consequences, such as extreme weather events, and wildfires, and also increased temperature [9]. Accordingly, temperature-related diseases, including violent behavior, are matters of increasing public health significance.

To the best of our knowledge, no study has evaluated the association between temperature and crime events in Hanoi, Vietnam. Vietnam is a Southeast Asian nation, a region at high risk for climate change-related hazards like rising temperature [10].” (Lines 48-63)

In the Discussion, we also mentioned our innovations that: “To our knowledge, this is the first study to investigate an association between daily temperature and crime risk in Vietnam.”

Lines 267 – 280, we compare our results to other studies. Furthermore, there is less research investigating the temperature-crime association, compared to the associations between temperature and diseases, such as cardiovascular diseases. Therefore, the effects of temperature on crime need more attention. In our opinion, this emphasis of a unique outcome associated with temperature in a relatively understudied study region is the true scientific benefit of this research. Our innovation lies in this exposure-outcome relationship, as opposed to the methods, which as the Reviewer pointed out are fairly well established.

Thank you for your consideration of this manuscript.

Sincerely,

Bruce H. Alexander, PhD
Professor and Head

Mayo Professor in Public Health

Reviewer 2 Report

Some questions for the authors:

Lines 11 and 12: the relation temperature and crime is not well understand, especially in developing countries. It would be interesting adding more studies from these countries:

India: https://www.researchgate.net/profile/Anita-Mandal/publication/343850659_Ambient_Temperature_and_Crime_in_Bihar/links/5f50fe21a6fdcc9879c52829/Ambient-Temperature-and-Crime-in-Bihar.pdf

Irán: doi:10.22131/sepehr.2020.47888

Colombia: https://doi.org/10.1177/0013916519878213

Brazil: https://doi.org/10.1016/j.worlddev.2022.105933

With a simple view in acedemic google, you can get references for USA, South Africa, Greece, China or Russia.

Lines from 28th to 35th: following a general trend in many studies, the authors include a reference to climatic change. I don't understand why. At no part of the paper, you are researching the evolution of temperatures or a possible increase of crime events. Only, effects of daily temperature on crime events. That is not a climatic change study. My recommendation is not including this paragraph. Curiously, the highest number of crimes was in 2014 (line 165 and S2) and from then, there is a great negative trend. NO problem with the climatic change.

You insist again in lines 260 and 261. I insist again with the same recomendation: erase these lines. Climatic change forecasts higher temperatures. Do you want to make out that with climatic change Hanoi will have more crimes? No. Crimes decrease from 2014.

And climatic change again between 267th and 279, with references of daily temperatures. No more comment about that. I think I have been enough clear about this item.

Lines 84 and 85: what is the reason for not collecting the drug-related and prostitution crimes?

Lines 128, 137 and 138: method to calculate RRI. RRI is defined as relative risk. It is calculated "(relative risk -1) x 100". Relative risk is used to calculate relative risk. I don't understand it. The method must be described more clearly.

Lines 149 and 150: Why do you use the first percentile to center the curve?

Lines 151 and 152: What R packages have been used?

Results: An important part of the results is used to describe tables and figures not printed in the paper, only in supplementary material. I think it is very important to include table S4 with the AIC values. Specially, I don't understand the differences between tables 2 and S4. In the first, you describe the models, from 0 to 4. But these models do not appear in table S4. From which model are data in table S4?

In table 2, you show some numbers in brackets. If you show, they'll be important, but you don't say what these numbers are. Confidence intervals?

You only mention briefly the seasonal evolution of crimes. You don't try to explain because months with more crimes are not the hotter ones. Any relation with the decrease of events with temperatures higher than 35 degrees.

Lines 265 and 266: Why do not include the reasons for lower crime numbers in Hanoi, with higher temperatures?

Lines 290 and 291: you don't describe the causes of higher crime number with higher temperatures. It is not a colateral question. It is a main question: the justification of the statistical relation you are researching.

Author Response

Dear Reviewer,

We thank the Reviewer for your comments regarding our manuscript. We have made several changes based on the Reviewer’ comments to improve the manuscript.

Lines 11 and 12: the relation temperature and crime is not well understand, especially in developing countries. It would be interesting adding more studies from these countries:

India: https://www.researchgate.net/profile/Anita-Mandal/publication/343850659_Ambient_Temperature_and_Crime_in_Bihar/links/5f50fe21a6fdcc9879c52829/Ambient-Temperature-and-Crime-in-Bihar.pdf

Irán: doi:10.22131/sepehr.2020.47888

Colombia: https://doi.org/10.1177/0013916519878213

Brazil: https://doi.org/10.1016/j.worlddev.2022.105933

With a simple view in acedemic google, you can get references for USA, South Africa, Greece, China or Russia.

We thank the Reviewer for this suggestion and have added more studies in developing countries to improve the introduction. However, we did not cite the first two suggested papers as both papers focus on season of the year and not explicitly temperature.  The following paragraph, which starts on line 40, was edited to include other references

“Little research has been done in developing countries to examine the effects of temperature on crime, and no study has been done in Vietnam. Gates et al., found an increase in homicides for each 1 °C in maximum temperature in South Africa[6]. However, Trujillo et al., indicated that each 1 °C in maximum temperature is associated with 0.8% increase in interpersonal violence but decrease in homicides in Colombia[7]. Existing research found that adverse weather shocks in the form of droughts lead to an increase in violent crime in rural Brazil[8]. These studies indicate that ambient temperature is associated with crime rates, but different regions may show different relationships.” (Lines 48-55)

Lines from 28th to 35th: following a general trend in many studies, the authors include a reference to climatic change. I don't understand why. At no part of the paper, you are researching the evolution of temperatures or a possible increase of crime events. Only, effects of daily temperature on crime events. That is not a climatic change study. My recommendation is not including this paragraph. Curiously, the highest number of crimes was in 2014 (line 165 and S2) and from then, there is a great negative trend. NO problem with the climatic change.

You insist again in lines 260 and 261. I insist again with the same recomendation: erase these lines. Climatic change forecasts higher temperatures. Do you want to make out that with climatic change Hanoi will have more crimes? No. Crimes decrease from 2014.

And climatic change again between 267th and 279, with references of daily temperatures. No more comment about that. I think I have been enough clear about this item.

We agree with Reviewer that climate change is not primary question of interest in this study. To focus on the effects of temperature on human health, we re-arranged the Introduction, moving the second paragraph to be first and emphasized the health effects of temperature. However, we believe that rising temperature is also important in the context of climate change, particularly in low- and middle-income nations where populations are anticipated to be disproportionately impacted by climate effects, so we included a paragraph (third) to make this point.  Between lines 260 -261 and 267- 279, we agree with the Reviewer, and we removed the lines 260 – 261 and the statement: “In the context of climate change” (Line 297)

“Ambient temperature is linked to several human health effects, including communicable and non-communicable diseases, and behavior changes.” (Lines 36,37)

“Related to this problem is the larger issue of climate change, which poses significant public health concerns, threatening humanity through the perpetuation of adverse environmental consequences, such as extreme weather events, and wildfires, and also increased temperature [9]. Accordingly, temperature-related diseases, including violent behavior, are matters of increasing public health significance” (Lines 56 – 60)

Lines 84 and 85: what is the reason for not collecting the drug-related and prostitution crimes?

We thank the Reviewer for noting this as it needs clarification in the manuscript. Drug-related crimes and prostitution are a different class of crimes in the Vietnamese criminal code and not managed by the same unit that manages crimes against persons. We included the following statement lines 99-102 and stated this lack of data as a limitation of our study (Line 350-351).

“Other non-violent crimes that are not directed at individuals, such as drug-related and prostitution, were not included in this study because those types of crime were not reported and managed by the PC02 Department.” (Lines 99- 102)

“Additionally, some crimes such as drug use and prostitution are not reported by the PC02 Department with other violent/non-violent crime types.” (Lines 351 – 352)

Lines 128, 137 and 138: method to calculate RRI. RRI is defined as relative risk. It is calculated "(relative risk -1) x 100". Relative risk is used to calculate relative risk. I don't understand it. The method must be described more clearly.

We agree that the use of this metric could use some clarification. We report the results of percent relative risk increase (RRI) to describe the change in risk associated with a temperature change as a percent increase or decrease. We have included the following clarification to the manuscript.

The interpretation of the RRI can be considered the percent relative effect and is directly derived from the relative risk. Reporting the RRI allows a change in risk that may be lower in magnitude, but still statistically significant, to be more easily interpreted. It is calculated as the (Relative Risk -1) x 100[4]” (Lines 148- 151)

Lines 149 and 150: Why do you use the first percentile to center the curve?

We agree with the Reviewer that it is important to explain our choice of the first percentile as a reference point. Our objective was to show a non-linear change in risk from a low baseline temperature through the distribution of temperature exposures warmer than this point. Temperature and health studies in non-Vietnam studies will usually center around 0 degrees or freezing, but since Hanoi doesn’t ever reach that point, we selected a suitable low temperature instead. We clarify this in the text

The curve was centered at 12 °C, which represent the 1st percentile of daily mean temperature for the entire 7 years to demonstrate the change in risk from a low temperature exposure point. The minimum recorded temperature was not selected to reduce the effects of extreme outliers in the final model.” (Lines 171-173)

Lines 151 and 152: What R packages have been used?

We thank the Reviewer for this suggestion to improve our manuscript. We added text to highlight the packages

“relying on the dlnm package for statistical assessments[12]” (Lines 175-176)

Results: An important part of the results is used to describe tables and figures not printed in the paper, only in supplementary material. I think it is very important to include table S4 with the AIC values. Specially, I don't understand the differences between tables 2 and S4. In the first, you describe the models, from 0 to 4. But these models do not appear in table S4. From which model are data in table S4?

We agree with the Reviewer that the decision to use the results for the mean daily temperature can be more clearly explained. The results from Table 2 are the primary results and show slightly different results for minimum, mean, and maximum temperature. We believe the mean values are more representative. The results for the AIC values for Model 1 are presented in table S4 as supplemental evidence for emphasizing the results using the daily mean temperature values. The data from Table S4 are not essential to interpreting the results we chose to keep this as a supplement. We have revised this paragraph (Lines 214-219) to clarify this.

In table 2, you show some numbers in brackets. If you show, they'll be important, but you don't say what these numbers are. Confidence intervals?

We thank the Reviewer for this comment. We made changes in text in Table 2 to make the numbers easier to read.

Model 0**

Model 1+

Model 2±

Model 3

Model 4§

RRI

(95%CI)

RRI

(95%CI)

RRI

(95%CI)

RRI

(95%CI)

RRI

(95%CI)

Violent Crime

Minimum

7.8

(-2.5; 19.1)

7.1

(-3.9; 18.3)

4.0

(-7.7; 15.5)

9.9

(-1.3; 22.5)

7.7

(-4.0; 20.9)

Mean

10.4 *

(0.7; 21.1)

9.9 *

(0.2; 20.5)

7.9

(-2; 18.9)

11.7 *

(1.2; 23.2)

10.2

(-0.8; 22.5)

Maximum

8.1 *

(0.6; 16.2)

7.9 *

(0.4; 15.9)

6.2

(-2.2; 14.8)

8.9 *

(0.9; 17.5)

7.2

(-1.4; 16.6)

Non-violent Crime

Minimum

5.9

(-0.1; 13.1)

5.6

(-1.0; 12.7)

6.3

(-0.1; 14.1)

7.0

(-0.4; 14.9)

8.4 *

(0.3; 17.2)

Mean

7.08 *

(0.7; 13.7)

6.8 *

(0.6; 13.5)

7.7 *

(1.0; 15.0)

8.1 *

(1.3; 15.4)

10.0 *

(2.5; 18.2)

Maximum

5.05*

(0.2; 10.1)

5.0*

(0.2; 10.1)

5.9*

(0.6; 11.6)

5.6*

(0.4; 11.1)

7.4*

(1.5; 13.6)

All Crime types

Minimum

6.3 *

(0.6; 12.2)

5.9 *

(0.3; 11.8)

5.6

(-0.1; 12.0)

7.6 *

(1.4; 14.2)

8.2 *

(1.4; 15.4)

Mean

7.8 *

(2.5; 13.3)

7.5 *

(2.3; 13.2)

7.8 *

(2.1; 13.7)

8.9 *

(3.2; 15.0)

10.1 *

(3.7; 16.8)

Maximum

5.8 *

(1.7; 10.1)

5.7 *

(1.7; 10.0)

6.0 *

(1.5; 10.7)

6.5 *

(2.0; 11.0)

7.3 *

(2.4; 12.5)

Missing data (%)

32

(1.25%)

32

(1.25%)

281

(10.99%)

309

(12.10%)

544

(21.28%)

You only mention briefly the seasonal evolution of crimes. You don't try to explain because months with more crimes are not the hotter ones. Any relation with the decrease of events with temperatures higher than 35 degrees.

Lines 265 and 266: Why do not include the reasons for lower crime numbers in Hanoi, with higher temperatures?

We thank the Reviewer for these comments. We agree that temperature-crime associations are complex. In this study, we examined crime risk of each 5 °C increase in daily mean temperature and not focus on seasonal evaluation of crime, but we did adjust for season to control the effects of season on crime rates and temperature variations.

We explained the reason for decrease crime risk at high temperatures in Discussion section, and this trend was observed in previous studies

“It can be explained by the Social Escape or Avoidance Theory and the Negative Affect Escape Model[20][21], and supported in  previous observational studies [4][17]. At very high temperatures, people will reduce social interaction to avoid heat, so crime risk may decrease.” (Lines 321- 324)

Lines 290 and 291: you don't describe the causes of higher crime number with higher temperatures. It is not a colateral question. It is a main question: the justification of the statistical relation you are researching.

We agree with the Reviewer that the temperature-crime association is the main question. In Lines 290-291, we discussed the associations with temperature followed an inverted U-shaped response with a decrease crime risk at high temperature (above 30 °C) for non-linear effect models. As described above, we offered potential theory for these effects (Lines 321- 324) which also seen in others research, whereby extreme heat events will result in less interaction and a reduced likelihood crime.

Thank you for your consideration of this manuscript. Please let me know if you have any concern regarding to the manuscript

Sincerely,

Bruce H. Alexander, PhD
Professor and Head

Mayo Professor in Public Health

Round 2

Author Response

Dear Reviewer,

We thank the Reviewer for their comments regarding the second version of our manuscript. We have made several changes and clarifications in response to the Reviewer’s comments.

Firstly, there is no literature cited in the introduction to support the relationship between violence and non-violence and crime. Other studies show the relationship between violence crime and non-violence crime, violent crime and temperature. Is there any literature support for the relationship between non-violent crime and temperature?

We agree that the literature supporting an association between temperature and non-violent crime could be more clearly described in the introduction section. To clarify, we made the following changes to provide evidence to support this hypothesis more explicitly.

“Research across 436 U.S. counties estimated that each 10 °C increase in daily temperature was associated with an 11.92% increase in the risk of violent crime and 6.14% increase in the risk of non-violent crime” (Lines 37-38)

“A single-city study in China showed the positive associations between temperature and both violent and non-violent crimes” (Lines 49-50)

Secondly, why does the explanatory variable of temperature distinguish between daily minimum, daily mean and daily maximum? When the temperature rises by 5 degrees Celsius, these temperature types will cross each other. So, there are problems in the selection of these variables.

The minimum, maximum, and mean temperature for a single day are distinct, although correlated, measures with the mean reflecting overall exposure over the course of a day and minimum and maximum reflecting exposure at a short interval. The following identifies how this is described in the paper. In section 2.3 (Lines 111-124), we described how to calculate each daily temperature measurement based on the hourly temperature for each day, which indicates they are distinct measures. In the linear models, we clarify that we examined the effects of each temperature measurement and crime type separately (Lines 148- 149). We estimated the risk of day-to-day variation in crime counts and day-to-day variation of minimum temperature, mean, or maximum temperature. Models were fit separately for each temperature measurement. We did not put all three temperature measurements in a single model, as their purpose was to demonstrate robustness in our temperature and crime association.

“All models were also fit separately for each temperature measurement and crime outcome: violent crime, non-violent crime, and the combination of violent and non-violent crime” (Lines 148- 149)

Third, the impact of multivariate in the model on crime was not analyzed

We are not quite sure what the reviewer is referring to with respect to the impact of multivariate in the model on crime. For this manuscript, we analyzed data at a population-level study, not an individual-level study. Thus, the impact of individual factors, such as age and gender, were not available for analysis.  (Lines 329- 333). Second, we controlled for long-term trends by year, season, and public holidays, as well as relative humidity and PM2.5 as a priori confounders in the linear models. However, due to the observed a high percentage of missing data for atmospheric condition data, other than temperature, we did not control for those variables in the non-linear models.

Finally, why the mean daily minimum, mean, maximum temperatures reflect the same crime occurrences in Table 1

We are not entirely clear what the reviewer is questioning.  Table 1 describes the data from 2013-2019 that was the basis for the analysis. The minimum column shows the lowest temperature and the minimum number of crime counts observed on a day in that period; the mean is the average temperature and daily crime counts in that period; the maximum is the highest temperature observed and greatest number of crimes on any single day in that period. We want to re-iterate that these exposures were evaluated separately to demonstrate robustness in the crime and temperature response relationship and we hypothesized similar qualitative relationships between these distinct exposure measures.

We hope these responses clarify uncertainties about the manuscript and we thank you again for considering this manuscript.

Sincerely,

Bruce H. Alexander, PhD
Professor and Head

Mayo Professor in Public Health